# Photoexcitation of Ge_9_^−^ Clusters in THF: New Insights into the Ultrafast Relaxation Dynamics and the Influence of the Cation

**DOI:** 10.3390/molecules25112639

**Published:** 2020-06-05

**Authors:** Nadine C. Michenfelder, Christian Gienger, Melina Dilanas, Andreas Schnepf, Andreas-Neil Unterreiner

**Affiliations:** 1Institut für Physikalische Chemie, Karlsruher Institut für Technologie (KIT), Kaiserstr. 12, 76131 Karlsruhe, Germany; nadine.michenfelder@kit.edu (N.C.M.); uxffr@student.kit.edu (M.D.); 2Institut für Anorganische Chemie, Universität Tübingen, Auf der Morgenstelle 18, 72076 Tübingen, Germany; christian.gienger@uni-tuebingen.de

**Keywords:** ultrafast dynamics, metalloid cluster, germanium, photoexcitation, excess charge

## Abstract

We present a comprehensive femtosecond (fs) transient absorption study of the [Ge_9_(Hyp)_3_]^−^ (Hyp = Si(SiMe_3_)_3_) cluster solvated in tetrahydrofuran (THF) with special emphasis on intra- and intermolecular charge transfer mechanisms which can be tuned by exchange of the counterion and by dimerization of the cluster. The examination of the visible and the near infrared (NIR) spectral range reveals four different processes of cluster dynamics after UV (267/258 nm) photoexcitation related to charge transfer to solvent and localized excited states in the cluster. The resulting transient absorption is mainly observed in the NIR region. In the UV-Vis range transient absorption of the (neutral) cluster core with similar dynamics can be observed. By transferring concepts of: (i) charge transfer to the solvent known from solvated Na^−^ in THF and (ii) charge transfer in bulk-like materials on metalloid cluster systems containing [Ge_9_(Hyp)_3_]^−^ moieties, we can nicely interpret the experimental findings for the different compounds. The first process occurs on a fs timescale and is attributed to localization of the excited electron in the quasi-conduction band/excited state which competes with a charge transfer to the solvent. The latter leads to an excess electron initially located in the vicinity of the parent cluster within the same solvent shell. In a second step, it can recombine with the cluster core with time constants in the picosecond (ps) timescale. Some electrons can escape the influence of the cluster leading to a solvated electron or after interaction with a cation to a contact pair both with lifetimes exceeding our experimentally accessible time window of 1 nanosecond (ns). An additional time constant on a tens of ps timescale is pronounced in the UV-Vis range which can be attributed to the recombination rate of the excited state or quasi conduction band of Ge_9_^−^. In the dimer, the excess electron cannot escape the molecule due to strong trapping by the Zn cation that links the two cluster cores.

Academic Editors: Constantina Papatriantafyllopoulou and George E. Kostakis

## 1. Introduction

The observation/tracking of electrons in photoexcited molecules, nanoclusters or particles is a very interesting research field in chemistry involving intra- and intermolecular electron transfer, conduction mechanisms or magnetic properties of molecules with regard to photochemical reactions and photocatalysis [1,2,3,4,5,6,7,8,9,10]. Transient absorption spectroscopy is a versatile tool to investigate photophysical and chemical mechanisms and processes, for example (bi-)radical lifetimes [11] or electron transfer [12]. Within inorganic chemistry, it was also used to understand the redox photochemistry of a molybdenum(II) cluster [13] or to find appropriate building units for photoactive metal-organic-frameworks (MOFs) [14]. Charged compounds offer good access via charge transfer transitions to their electronic structure in the excited state [15]. A charge transfer can occur intra- or intermolecularly or as a charge transfer to the solvent, when working in solution [16,17]. Most excited inorganic clusters and complexes show intramolecular charge transfer [8,12,18,19] or triplet state dynamics [20,21,22,23,24]. The metalloid cluster [Ge_9_(Hyp)_3_]^−^ (Hyp = Si(SiMe_3_)_3_) [25] (see Figure 1) with its high-yield synthesis, high stability under inert conditions and good solubility in organic solvents was used for, among others, build up reactions which give access to cluster aggregates and chain compounds with definite compositions [25,26,27,28,29,30,31,32,33]. The Ge_9_^−^ entity can be linked, for example, by transition metal ions like Cu^+^, Ag^+^, Au^+^ [26] or Zn^2+^, Cd^2+^, Hg^2+^ [27] to build up a neutral dimer in the latter case. The chemical variety of this compound opens up a large playground to understand its chemical and physical properties. This cluster differs from most inorganic clusters as it can probably completely abstract an electron from the cluster core [2] which allows both intra- and intermolecular charge transfer. This approach gives access to different photo induced reactions simultaneously. To observe the processes after photoexcitation, e.g., the detachment of an electron from the cluster, fs transient absorption (TA) spectroscopy was applied. In solution, such an abstracted electron can dissolve as a so-called solvated electron [34]. In presence of a cation a contact pair with the solvated electron forms [35,36]. The well-known absorption spectra of the solvated electron (λ_max_ = 2120 nm) [37] and the cation-electron contact pairs (e.g., [Na^+^,e^−^]_THF_: (λ_max_ = 890 nm) [38], [Li^+^,e^−^]_THF_ (λ_max_ = 1180 nm) [39] and [K^+^,e^−^]_THF_ (λ_max_ = 1125 nm) [40]) peak in the NIR spectral range. The Ge_9_^−^ cluster moiety can be excited in the UV-Vis range as presented in reference [2]. With single wavelength probing in the near infrared (NIR) region, indications for excess electron formation were found whereas in the UV-Vis regime cluster dynamics were observed [2]. In another publication, we also investigated a neutral complex [Ge_9_(Hyp)_3_FeCp(CO)_2_] which gave new insights into ultrafast dynamics of this cluster-type [5]. Surprisingly, no long-lived excited states exceeding lifetimes of more than several hundreds of picoseconds were observed, which was attributed to the special role of cationic iron.

In this contribution, we present a comprehensive study on the [Ge_9_(Hyp)_3_]^−^ cluster dynamics after photoexcitation with special emphasis on the influence of different counter ions like K(crypt)^+^ (crypt = [2.2.2]cryptant) (K(crypt)Ge_9_) in comparison to Li^+^ (LiGe_9_) or K^+^ (KGe_9_). Complementing studies on [(Hyp)_3_Ge_9_ZnGe_9_(Hyp)_3_] [27,33] (Ge_9_ZnGe_9_) show a strong influence of dimerization on the long timescale dynamics of the cluster which is crucial for the understanding of possible relaxation pathways. This study also includes results obtained from a new experimental setup [5,41] that now allows long timescale (1 ns) broadband probe in the UV-Vis as well as in the NIR spectral range. This should allow the observation of at least a section of the high-energy tail of the solvated electron spectrum in THF, which still peaks (ca. 2100 nm) beyond our experimentally accessible range of 1400 nm. The analysis results in a concept for the charge transfer processes after photoexcitation of the metalloid [Ge_9_(Hyp)_3_]^−^ cluster and the role of counter ions and dimerization.

## 2. Results and Discussion

### 2.1. Steady State Spectroscopy in Solution

Stationary absorption spectra for K(crypt)Ge_9_, Ge_9_ZnGe_9_, LiGe_9_ and KGe_9_ are shown in Figure 2 and Appendix A. For the monomers, a spectral broad absorption band in the UV spectral region was observed, which extends into the Vis regime. No absorption bands could be found in the near infrared (NIR) spectral range. The results nicely match the known spectrum of the LiGe_9_ cluster [2], whereby the spectrum does not significantly differ with K^+^ or K(crypt)^+^ as counterion. As known from time dependent density functional theory (TD-DFT) calculations using a grid-based projector-augmented wave code [2], the transitions in the visible region primarily originate from atomic orbital contributions of the Ge_9_^−^ cluster core. For the Ge_9_ZnGe_9_ cluster, enhanced bands around 390 nm and 490 nm were observed [33]. Known from TD-DFT calculations in the gas phase, the transitions from the HOMO*−*1 to the LUMO were found at 500 nm. Around 400 nm multiple transitions from the HOMO*−*4, HOMO*−*3 and HOMO*−*5 to the LUMO+2, LUMO+3, LUMO+4 and LUMO+5 contribute to the peak [33].

### 2.2. Transient Absorption Spectroscopy in Solution

#### 2.2.1. Dynamics of the Monomer

After excitation into the UV band (267 nm) of K(crypt)Ge_9_, where K^+^ is trapped by a [2.2.2]cryptand, TA spectra as shown in Figure 3 were obtained. For this compound, a spectral broad TA band extending from the visible to the NIR range was observed. In the UV region, it is superimposed by the ground state bleach (GSB), which leads to negative transient response around 400 nm. Ground state population does not recover within the experimental time window (1 ns). Comparably long-lived absorption bands dominate throughout the blue to the NIR spectral region. The transient response on a longer timescale shows a steady increase from visible to NIR, which can be interpreted as a blue tailing end of an absorption band that will be discussed below. 

This can be further visualized by inspecting single transients at different wavelengths (see Figure 4). Due to different techniques, the time resolution of the UV-Vis excited experiments is considerably shorter than in the NIR region as observable by a faster rise of the corresponding transients on an ultrashort timescale. The transients for the highest wavelengths tend to be among those with highest induced absorption while showing a continuous increase in the NIR on a long timescale.

To get a closer look on the dynamics and time constants of the processes, global analysis was done with an in-house written program in the MATLAB environment. The transients were fitted (see SI *Fit function)* with four time constants, which were essential to reproduce the data adequately and are given in Table 1 for all compounds and the corresponding amplitudes are presented as decay associated spectra (DAS) (see Figure 5 for K(crypt)Ge_9_). The DAS show the absorption spectra of the excited states, which correspond to the respective time constant.

The short timescale dynamics, which are pronounced mainly in the visible part of the spectrum can be attributed to the cluster core dynamics [2]. A more detailed analysis is challenging as suitable reference systems are unknown and theoretical calculations do not provide dynamic information. On the other hand, absorption bands of long-lived species in the NIR are often triplet states or charge transfer species like solvated electrons. Triplet states in this system are unlikely to be populated as previous calculations did not show energies of triplet states that may be accessible under our experimental conditions [2]. Therefore, we concentrate on the excess charge dynamics as potential candidates. The long-lived process (A_4_, τ_4_) can be mainly observed as GSB around 400 nm and as induced absorption in the NIR spectral region. It is known from gas phase experiments that the vertical electron detachment energy is 3.37 eV [2], which somewhat lowers in solution [42,43] eventually leading to a charge transfer to solvent (CTTS). Therefore, we compared the absorption spectra of the solvated electron in THF [37] with the long-lived DAS (A_4_) of the Ge_9_^−^-cluster (see the orange dashed curve in Figure 5). The match is within the error tolerance.

As a result of this agreement, we assume a formation of a solvated electron in THF solution that was abstracted from the excited Ge_9_^−^ cluster after photoexcitation. To understand the underlying processes, we apply a mechanism which was initially proposed by Barthel et al. [16] to describe dynamics after photoexcitation of Na^−^ in THF solution. In this scenario, the excited Na^−^* can undergo a CTTS reaction within 0.7 ps. The resulting excess electron is initially located in the vicinity of the parent Na^0^, leading to absorption bands of Na^0^ (peaking near 890 nm [16,38,44]) and the solvated electron (e^−^_solv_, 2120 nm in THF solution [37]). Partially, geminate recombination is possible that results in a decay of the Na^0^ and the e^−^_solv_ bands on a timescale of 1.5 ps. The remaining excess electrons can escape the parent Na^0^ to survive some ns until further recombination reaction occurs. 

Our system is of course much larger involving more degrees of freedom, but nevertheless there are some similarities. The following chemical equations show the adapted mechanism from Na^−^ [16] onto the Ge_9_^−^ cluster:(1)$$\begin{array}{l} \left[Ge_{9}^{-}\right]\overset{h\nu }{\to }{\left[Ge_{9}^{-}\right]}^{*}\\ {\left[Ge_{9}^{-}\right]}^{*}\overset{CTTS}{\to }\left[Ge_{9}^{0}\right]\cdot e_{solv}^{-}\\ \left[Ge_{9}^{0}\right]\cdot e_{solv}^{-}\overset{recomb.}{\to }\left[Ge_{9}^{-}\right]\\ \left[Ge_{9}^{0}\right]\cdot e_{solv}^{-}\overset{\mathrm{escape}}{\to }\left[Ge_{9}^{0}\right]+e_{solv}^{-} \end{array}$$

The negatively charged cluster with its relatively low electron detachment energy releases an electron after photoexcitation. Subsequently, this electron is transferred to the solvent (CTTS) which can be observed as an ultrafast decay of the excited Ge_9_^−^ species with time constant τ_1_. In line with earlier studies on a LiGe_9_ cluster [2,45], vibrational relaxation probably occurs on this timescale, too. The second decay process (τ_2_) can be attributed to the recombination with the parent Ge_9_^0^ cluster of the solvated electrons localized nearby. 

The third time constant in this simple model has no relating process. We will return to this issue in the further course of the Discussion section. 

Solvated electrons that escape the interaction area can survive longer than our experimental delay range (1.2 ns) [37]. For K(crypt)Ge_9_ the absorption spectrum after 1 ns and the decay associated absorption spectrum of τ_4_ matches the high energy tail of the solvated electron absorption spectrum in THF with its maximum at 2120 nm [37] (see SI Appendix A).

#### 2.2.2. Impact of the Counter Ion

Next, we concentrate on the influence of different counterions. It is known, that the cation (K^+^) can be bound to the cluster core [46] if not trapped by crypt. Therefore, it can be assumed for KGe_9_ and LiGe_9_ that the distance between the cluster core and the cation is small enough for an interaction to take place, which is expected to influence the relaxation dynamics. At first glance, the transient spectra of KGe_9_ and LiGe_9_ (see Appendix A) are similar to the one of K(crypt)Ge_9_ showing a spectral broad long-lived transient absorption throughout the entire experimentally observed spectral range. A closer look, however, reveals subtle differences. For example, time constants from global fitting as described above are in the same order of magnitude except for τ_3_, which is much shorter for the free cation species (Table 1). In addition, the longest timescale, designated with time constant τ_4_, shows a somewhat different spectral signature, which does not have much similarity with the solvated electron spectrum in THF. While the UV-Vis range again shows typical spectral features assigned to cluster dynamics, there is a small, but reproducible maximum around 1050 nm in the DAS (Figure 6). 

This feature nicely matches another form of charge transfer species, which can be attributed to a contact pair between an electron and K^+^ [40]. Again, we apply concepts from suitable reference systems. For example, it is known that I^−^ releases an electron which forms a contact pair with Na^+^, i.e., [Na^+^,e^−^], in aqueous NaI [1]. The corresponding contact pair absorption maxima for Li^+^ and K^+^ lie at 1180 [39] and 1125 [40] nm (see also Appendix A).

The absorption spectrum of a contact pair of the electron with K(crypt)^+^ is not known but one would expect it at longer wavelengths as it should be weaker bound than the free ion contact pairs. Therefore, the spectra of the contact pairs and the solvated electron may superimpose. The formation process of the solvated electron and the cation-electron contact pair cannot be observed in our case. The cation has only weak influence on the short timescale dynamics of the cluster (τ_1_ and τ_2_), but on the third process and on the charge transfer species (τ_4_). The charge transfer species are a strong hint that photoexcitation of the Ge_9_^−^ species leads to an electron detachment and finally to a release of this electron to the solvent, which is in line with earlier analysis by our groups [2]. In this scenario, the low intensity band around 460 nm of *A_4_* for KGe_9_ is assigned to dynamics of a Ge_9_^0^ species. It remains from the electron detachment process. All transient response in this spectral region can be related to cluster dynamics of Ge_9_^0^ and unaltered Ge_9_^−^ moieties, i.e., clusters where photodetachment did not occur. 

#### 2.2.3. Dynamics of the Dimer

To further substantiate this analysis, excitation of the cluster dimers such as Ge_9_ZnGe_9_ is helpful. After excitations at 267/258 nm the transient spectra (see Figure 7) reveal some similar features as the monomer clusters and can be summarized as follows: (i) The broad TA bands from Vis to NIR are superimposed by the negative transient response of the GSB around 390 and 480 nm. (ii) Global analysis reveals four-time constants (see Table 1), which are still in the same order of magnitude as for the monomers. In contrast to the monomeric compounds, the long-lived transient response in the NIR does neither match the solvated electron spectrum in THF nor a cation/electron contact pair. Instead, the spectrum shows more similarity with the excited cluster core spectra such as there are stronger amplitudes towards the blue spectral part compared to red ones. Finally, we point out that the overall transient response almost completely recovers after our maximum delay time of 1 ns. As a result, this makes a CTTS less likely for the dimeric form. 

In addition to the UV excitation, the stationary absorption spectrum of Ge_9_ZnGe_9_ allows another excitation regime at 400/388 nm (3.09/3.20 eV). This is, of course, slightly lower than the photodetachment energy known from gas phase experiments. On the other hand, it is known for water [42] and THF [43] that the ionization/photodetachment energy decreases by almost 1 eV in solution compared to the gas phase making a photodetachment feasible. Indeed, the transient spectra (Appendix A) resemble the one after UV excitation as do the first three-time constants (see Table 1). The fourth time constant, however, is significantly shorter. 

So far, we attribute the short timescale processes τ_1_ to vibrational relaxation and CTTS and τ_2_ to geminate recombination processes for all compounds. For the monomer, the charge transfer leads to a solvated electron which can escape out of the interaction sphere of the parent cluster. It can combine with the counterion. Both leads to long lived TA in the NIR spectral range. For the dimer an abstraction of the electron cannot be observed. In the visible range, the transient response is attributed to the Ge_9_^−^ and Ge_9_^0^ species.

Finally, we address the interpretation of the third time constant τ_3_. The monomer with K(crypt)^+^ has a τ_3_ of 120 to 140 ps, whereas it decreases to 40 and 30 ps for the cations Li^+^ and K^+^, respectively. Indeed, the shortest value can be found for the K^+^ cation and for the cluster dimer (30 ps). An influence of the dimerization on the third time constant is not present. This differs from the fourth process which is shorter for lower excitation energy and is directly linked to the charge transfer and therefore to the electron detachment process. In addition, the transient spectra of the dimer suggest that this process is independent of the excitation wavelength. One can conclude that the state, which we describe with τ_3_, is not involved into the charge transfer process. Nevertheless, the cation, especially the trapped cation, interacts with the process we are looking for by e.g. electron withdrawing effects. In all investigated compounds, the DAS (A_3_) are similar to the spectra, which can be attributed to the cluster in its excited state with or without the detached electron. To explain these experimental findings, we take into account another concept known for charge transfer processes after photoexcitation, which was applied to explain excited state dynamics of [Ge_9_(Hyp)_3_FeCp(CO)_2_] [5]. In this case, no solvated electron or long-lived states occur, but a trap state exists with a lifetime of ~150 ps that can be attributed to a charge transfer state. This is possible for Ge_9_ZnGe_9_ but is unlikely for the monomers as their counterion is not directly linked to the cluster and the contact pair bands are pronounced in the NIR spectral range. In bulk materials like α-Fe_2_O_3_ nanoparticles, thin films, colloidal solutions [47,48,49], or Fe_10_Ln_10_ clusters [8], photoexcitation into the quasi-conduction band leads to a charge transfer from oxygen to iron [17]. Three-time constants can be found on a sub ps, ps, and tens of ps timescale that are attributed to vibrational relaxation and trapping, geminate recombination, and relaxation from trap states, respectively. By analogy, on the timescale of the CTTS (τ_1_) a concurrent process is therefore the localization around the cluster core in a low excited state or lower edge of a quasi-conduction band. τ_3_ can be attributed to the relaxation from the quasi conduction band or the excited state. Therefore, the influence of the cation on τ_3_ is related to the change of the potential energy surface of the excited state due to interaction with the electrostatic field of the cation. In Figure 8 an overview on the possible processes after photoexcitation is given. 

This concept is also helpful for the understanding of the long time-scale dynamics of the cluster dimer. For 400 nm excitation τ_4_ amounts to only 400 ps and is significantly shorter than after 267 nm excitation. Interestingly, the DAS for this time constant are quite similar. The ground state completely recovers within the experimental time window of 1.2 ns (after 400 nm excitation). The long-lived transient absorption in the NIR does not match the solvated electron absorption spectrum, neither after UV nor after 400 nm excitation. As the cation is part of the metalloid cluster entity, a contact pair spectrum is, as expected, not observable. Therefore, no evidence for a release of the electron into the solvent can be found. It is still possible that because of a very low concentration, a small amount of solvated electrons may not be observable, but the complete recovery of the ground state bleach does not promote any species left after 1 ns. Consulting the concept from bulk-like materials and results from [Ge_9_(Hyp)_3_FeCp(CO)_2_] [5], a trap state resulting from a charge transfer between the cluster core and the Zn cation like in the monomer complex [Ge_9_(Hyp)_3_FeCp(CO)_2_] [5], may serve as an explanation why the electron cannot completely be released from the cluster. Indeed, its lifetime depends on excitation wavelength, i.e., on the excess energy deposited in the excited state. As we are not aware of systematic energy-depending studies on such systems, the wavelength dependent lifetimes are still reminiscent to studies of excess electrons in THF [43] or liquid ammonia [7,50] whereby excess energy in the excited state determines survival probability.

## 3. Materials and Methods 

### 3.1. Sample Preparation

All reactions were carried out under a nitrogen atmosphere using standard Schlenk techniques or under argon atmosphere in a glove box. Toluene, hexane, and THF were dried with sodium, acetonitrile was dried with phosphorus(V)oxide, and pentane was dried with calcium hydride and all were distilled prior to use. 

#### 3.1.1. K_4_Ge_9_

K (1.54 g, 39.39 mmol) and Ge (5.0 g, 68.84 mmol) were heated in an evacuated quartz glass vial to 650 °C with a ramp of 100 °C per hour and then held at 650 °C for 72 h. After cooling down to room temperature the vial was opened in a glove box and K_4_Ge_9_ was obtained as a grey solid and characterized via X-Ray powder diffraction on a Stadi-P (STOE, Darmstadt, Germany) powder diffractometer using germanium monochromated Cu-K_α1_ radiation (λ = 154.06 pm and a Mythen 1K detector (see Appendix A)).

#### 3.1.2. KGe_9_Hyp_3_

K_4_Ge_9_ (3.0 g, 3.07 mmol) was suspended in 150 ml of THF. ClSi[Si(CH_3_)_3_]_3_ (3.0 mL, 2.56 g, 9.056 mmol) was heated to 60 °C to melt and then added quickly via syringe to the suspension. The mixture is stirred at ambient temperature for three days and after that the solvent is removed in vacuo. The residue is washed twice with 100 mL of pentane and then extracted with THF. After removal of the solvent the product is obtained as an orange solid with a yield of 38%. KGe_9_Hyp_3_ was used NMR pure or crystallized from toluene at −30 °C in the shape of orange needles. ^1^H-NMR (C_6_D_6_, 300 MHz): δ 0.52 (s, 81H, SiMe_3_), ^13^C-NMR (C_6_D_6_, 62.9 MHz): δ 3.04 (SiMe_3_), ^29^Si-NMR (C_6_D_6_, 49.7 MHz, inept-nd): δ −8.48 (decet, Si(*Si*Me_3_)_3_, 2*J* Si-H = 6.5 Hz), −105.59 (s, *Si*(SiMe_3_)_3_).

#### 3.1.3. K(crypt-2,2,2)Ge_9_Hyp_3_

KGe_9_Hyp_3_ (500 mg, 0.35 mmol) was dissolved in acetonitrile and layered with hexane. [2,2,2]-cryptand was dissolved in hexane and added to the hexane layer. Orange crystals formed at room temperature at the interface. ^1^H-NMR (thf-*d*_8_, 300 MHz): δ 0.52 (s, 81H, SiMe_3_), ^13^C-NMR (thf-*d*_6_, 62.9 MHz): δ 3.04 (SiMe_3_), ^29^Si-NMR (thf-*d*_8_, 49.7 MHz, inept-nd): δ −8.48 (decet, Si(*Si*Me_3_)_3_, 2*J* Si-H = 6.5 Hz), −105.59 (s, *Si*(SiMe_3_)_3_).

#### 3.1.4. LiGe_9_Hyp_3_

KGe_9_Hyp_3_ (1.00 g, 0.7 mmol) and LiBr (61 mg, 0.7 mmol) were dissolved in THF at room temperature and stirred overnight. After removing the solvent in vacuo the product was washed with pentane and then extracted with THF. It was obtained in the form of red crystals at −30 °C with 95% yield. ^1^H-NMR (thf-*d*_8_, 300 MHz): δ 0.24 (s, 81H, SiMe_3_), ^13^C-NMR (thf-*d*_8_, 62.9 MHz): δ 3.19 (SiMe_3_), ^29^Si-NMR (thf-*d*_8_, 49.7 MHz, inept-nd): δ −9.85 pm (decet, Si(*Si*Me_3_)_3_, 2*J* Si-H = 6.6 Hz), −105.59 ppm (s, *Si*(SiMe_3_)_3_), ^7^Li-NMR (thf-*d*_8_, 116.64 MHz): δ 0.39 (s, Li).

#### 3.1.5. ZnGe_18_Hyp_6_

ZnCl_2_ (13 mg, 95 µmol) and KGe_9_Hyp_3_ (256 mg, 178 µmol) were separately dissolved in THF and cooled to -78 °C. While stirring the ZnCl_2_ solution was added dropwise to the KGe_9_Hyp_3_ solution and afterwards the reaction solution was slowly warmed to room temperature overnight. The solvent was then removed in vacuo and the product extracted with pentane. ZnGe_18_Hyp_6_ crystallizes in the form of dark red needles at −30 °C with 80% yield. ^1^H-NMR (C_6_D_6_, 300 MHz): δ 0.54 (s, 162H, SiMe_3_), ^13^C-NMR (C_6_D_6_, 62.9 MHz): δ 3.25 (SiMe_3_), ^29^Si-NMR (C_6_D_6_, 49.7 MHz, inept-nd): δ −9.59 (decet, Si(*Si*Me_3_)_3_, 2*J* Si-H = 6.6 Hz), −105.30 (s, *Si*(SiMe_3_)_3_).

Under exclusion of oxygen and water THF (abs.) was added as solvent for stationary and transient absorption measurements. Measurements were done in cuvettes optimized for the use under inert conditions.

### 3.2. Steady-State Spectroscopy in Solution

Absorption spectra were obtained with an UV/Vis/NIR spectrometer Cary 500 (Varian, Palo Alto, CA, USA) in THF as solvent in a wavelength range between 200 and 2000 nm. Spectra were measured at room temperature in cuvettes made of fused silica (Hellma, Nürnberg, Germany) with 1 mm optical path length. Concentrations are about 1 mmol/L.

### 3.3. Transient Absorption Spectroscopy in Solution

To obtain time resolved spectra in the UV-Vis wavelength regime, an experimental setup described elsewhere [41] was used. Briefly, one small part (2–3 µJ) of the 800 nm (Astrella (Coherent, Santa Clara, CA, USA), 7 mJ, 35 fs, repetition rate 1 kHz) laser output was focused into a movable 2 mm CaF_2_ crystal (nortus optronic GmbH, Wörth am Rhein, Germany) to generate a white-light continuum between 350 and 720 nm. After passing the sample, the white-light is refracted by a fused silica prism and recorded by a CCD Camera (Series 2000, Si Photodetector, Entwicklungsbüro Stresing, Berlin, Germany). Pump wavelengths at 400 and 267 nm were generated by second harmonic generation and sum frequency mixing of the 800 and 400 nm pulses in BBO crystals, respectively. The spot size in the sample was about 200 μm, which was more than twice the white-light spot size. Excitation energies were for all wavelengths 400 nJ per pulse. Delay of the pump pulse was managed by a computer-controlled translation stage (maximum delay ~1.2 ns, Thorlabs, Newton, NJ, USA), whereby every second pulse was blocked with an optical chopper (Thorlabs), resulting in spectra with and without excitation. Differentiation results in ΔA spectra with a time resolution better than 100 fs. Data were collected with an in-house written Labview program. 

For recording TA spectra in the NIR spectral range, as described in an earlier publication [5], a CPA 2210 (Clark-MXR, Dexter, MI, USA, 775 nm, repetition rate 1 kHz, 1.3 mJ, 150 fs) was used. One part of the 775 nm beam propagated through a computer-controlled translation stage (maximum delay range 1.4 ns, Physical Instruments PI, Karlsruhe, Germany) and was used to generate an NIR white-light continuum between 900 and 1600 nm in a YAG crystal (nortus Optronic GmbH). The intensity of the refracted pulse (SF10 prism) was collected with a CCD Camera (Series 2000, InGaAs Photodetector, Entwicklungsbüro Stresing). Data were processed by the same program like in the other system adapted to the needs for NIR detection. Pump pulses were generated by second (388 nm) and third (258 nm) harmonic generation of the fundamental laser wavelength. Excitation energies were in the range of 1 μJ per pulse and spot sizes in the sample around 500 μm which, again, corresponded roughly two times the white-light spot size. Longer pulse duration of the fundamental and group velocity mismatch between UV/Vis pump pulses and NIR white-light detection resulted in a time resolution of roughly 200 fs obtained by data analysis.

## 4. Conclusions

We investigated the relaxation channels after photoexcitation of the [Ge_9_(Hyp)_3_]^−^ clusters with different cations (K^+^, Li^+^ and K(crypt)^+^) and as dimer (bound through Zn^2+^) with fs pump probe broadband absorption spectroscopy in THF solution. In analogy to the charge transfer of bulk-like materials and charge transfer to solvent processes in Na^−^, we identified four different processes: Within 0.5 ps after photoexcitation with a 267 nm laser pulse an electron transfer to the solvent takes place, whereas localization in the quasi conduction band and vibrational relaxation occur on a similar timescale. The geminate recombination between the excess electron and the parent cluster occurs within about 2 ps. On a tens of ps timescale one can observe the relaxation from the quasi conduction band. Some excess electrons can escape the cluster influence, leading to solvated electrons and contact pairs with the cations that show characteristic spectra in the NIR with lifetimes longer one ns. For the dimer, the excess electron cannot escape the trapping field of the Zn atom. This leads to a charge transfer state, which decays on a shorter timescale depending on the excitation wavelength. Hence, the dimer shows a normal relaxation pathway for an inorganic cluster in opposition to the monomer, which can completely release an electron into the solvent, which can act as reducing agent. In future work, the influence of different bound cations in form of dimers like Ge_9_ZnGe_9_ or complexes like [Ge_9_(Hyp)_3_FeCp(CO)_2_] and on the other hand of different sizes of substituents of the cluster seem to be promising targets for investigations.

## Figures and Tables

**Figure 1 molecules-25-02639-f001:**
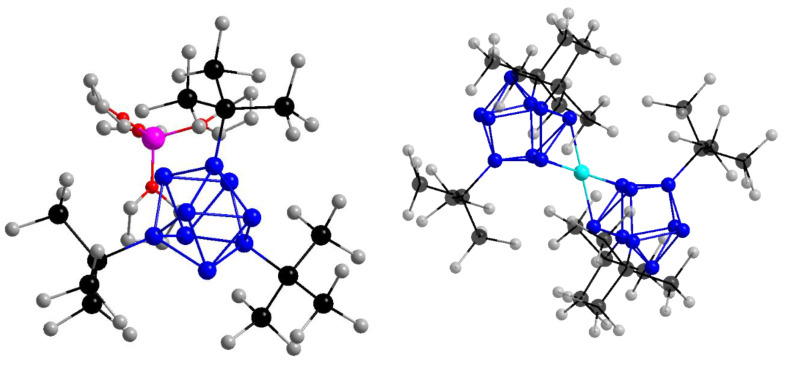
Molecular structure of LiGe_9_ (left) and Ge_9_ZnGe_9_ (right) from crystallographic data without hydrogen atoms. Color-code: Ge(blue); Zn(cyan); Si(black); C(grey); Li(pink); O(red).

**Figure 2 molecules-25-02639-f002:**
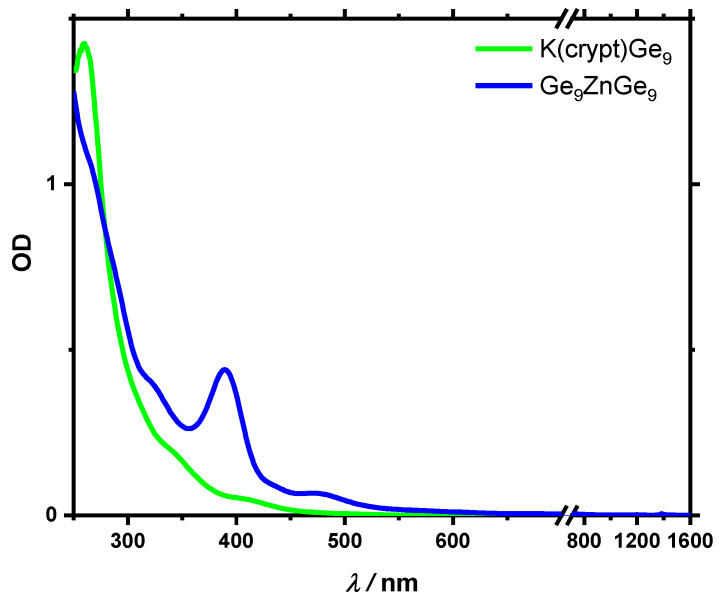
Absorption spectra for K(crypt)Ge_9_ and Ge_9_ZnGe_9_ between 250 and 1600 nm in THF solution with a concentration of about 1 mmol/L. Spectra of KGe_9_ and LiGe_9_ are very similar to K(crypt)Ge_9_ and can be found in the Appendix A.

**Figure 3 molecules-25-02639-f003:**
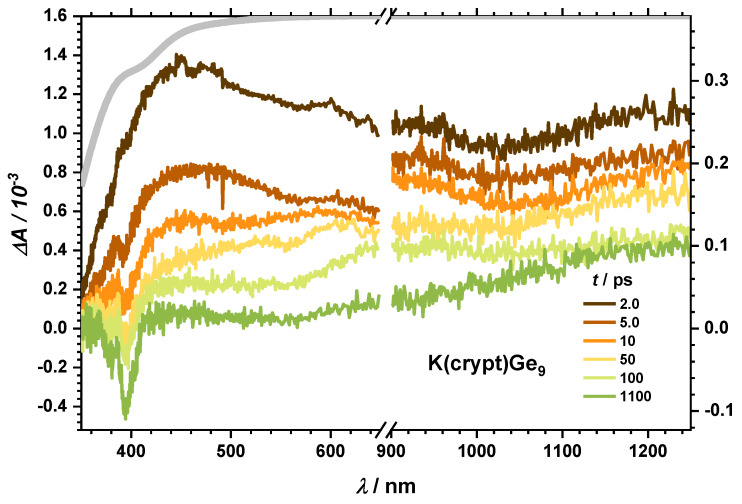
Broad transient absorption spectra for K(crypt)Ge_9_ in THF after excitation at 267/258 nm at given delay times. The gray line shows the ground state absorption spectrum with inverted OD axis. The lowered absolute transient response indicates absorption of the white-light probe beam by the sample between 350 and 390 nm.

**Figure 4 molecules-25-02639-f004:**
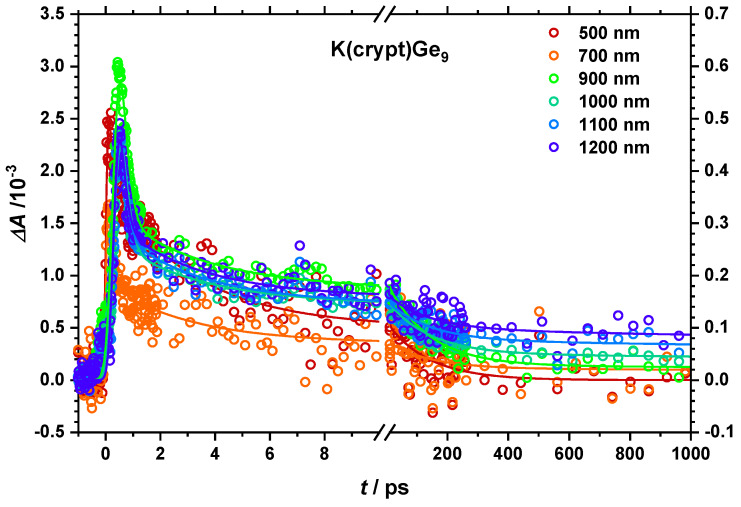
Transient response after 267/258 nm excitation at different wavelengths (circles) for K(crypt)Ge_9_^−^ in THF. On a long timescale one can see that the values continuously increase with longer wavelengths. Fit curves are displayed as lines.

**Figure 5 molecules-25-02639-f005:**
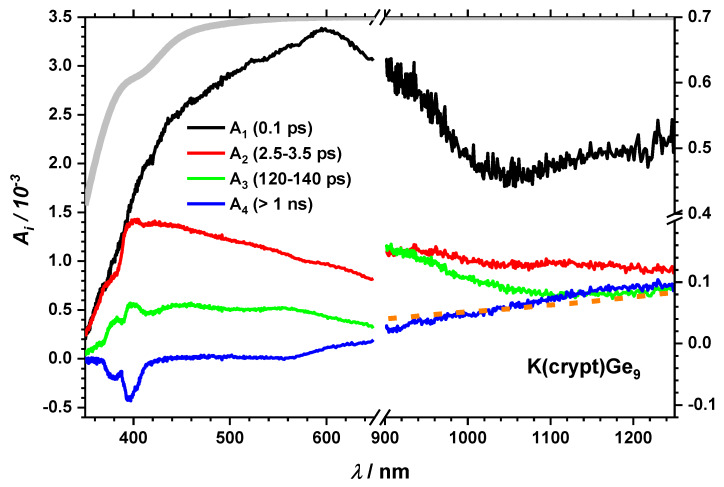
Decay associated spectra from global fit (further information is given in the Appendix A) for K(crypt)Ge_9_ in THF. A section of the absorption spectrum of the solvated electron in THF is given as an orange dashed line. Further spectra can be found in the Appendix A.

**Figure 6 molecules-25-02639-f006:**
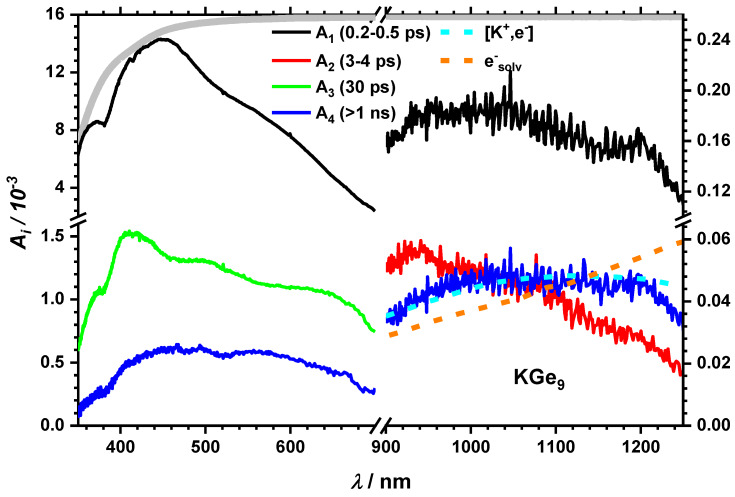
DAS after 267/258 nm excitation for KGe_9_ in THF. Dashed blue and orange lines show a [K^+^,e^−^] contact pair [40] and a section of the absorption of the solvated electron in THF [37]. The corresponding transient spectra and all spectra for LiGe_9_ can be found in Appendix A.

**Figure 7 molecules-25-02639-f007:**
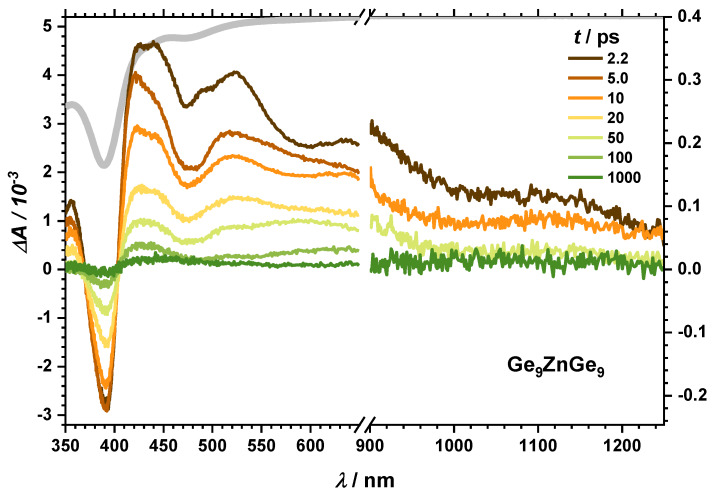
Transient spectra at given delay times for Ge_9_ZnGe_9_ in THF after 267 nm excitation. Ground state absorption spectrum in gray is inverted.

**Figure 8 molecules-25-02639-f008:**
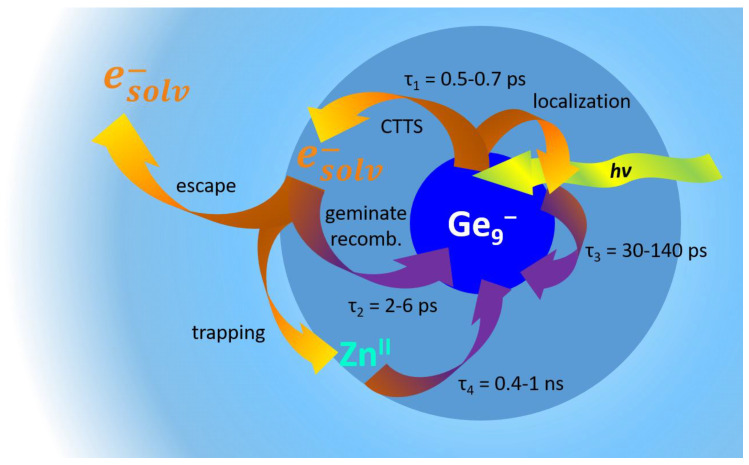
Illustration on the processes after photoexcitation (hν) of the Ge_9_^−^-cluster compounds. Recombination processes are depicted as violet arrows all other processes in orange.

**Table 1 molecules-25-02639-t001:** Time constants from global analysis after UV (267 nm) and additionally 400 nm excitation for Ge_9_ZnGe_9_.

	K(crypt)Ge_9_ (267 nm)	KGe_9_ (267 nm)	LiGe_9_ (267 nm)	Ge_9_ZnGe_9_ (267 nm)	Ge_9_ZnGe_9_ (400 nm)
τ_1_/ps	0.1	0.5	0.5−0.7	0.3	0.4
τ_2_/ps	2.5−3.5	3.3	2.4	2.5−5.8	1.8
τ_3_/ps	120−140	30−50	40−60	30	30
τ_4_/ps	>1000	>1000	>1000	700−1200	400

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
