# Peer review of "Photoexcitation of Ge9 Clusters in THF: New Insights into the Ultrafast Relaxation Dynamics and the Influence of the Cation"

_molecules, 2020, doi:10.3390/molecules25112639_

Round 1

Reviewer 1 Report

Authors present a complete fs transient absorption (TA) study of the [Ge9(Hyp)3] clusters solvated in THF with special attention on the role on different counter ions and a neutral cluster dimer with a central Zn atom. By looking carefully at UV/Vis and NIR TA changes with pump-probe delay in a 0 to 1 ns time range, they have evidenced 4 different relaxation regimes related to charge transfer or localized excited states. They were attributed to ultrafast charge transfer to solvent (CTTS), excited-state localization and vibration relaxation, germinal recombination followed by relaxation from quasi conduction band at intermediate timescale, and relaxation of a solvated electron at longer time scale. The study of cluster dimer has evidenced that excess electron cannot escape the trapping field of the Zn atom leading to the absence of long-time time relaxation contribution in TA spectra, while the presence of counter ions reveals the impact of the contact pair on the long lifetime dynamics.

I have no major opposition to the work presented and the conclusions stated in the manuscript. However, several suggestions to improve the article should be considered by the authors to help the reader along the text, before being considered for publication in Molecules – MDPI.

General comments:

  1. In abstract/introduction, I suggest them to be more specific on the interest to study ultrafast dynamics in such systems in this context and why comparing monomers with counter ions and neutral dimer to understand cluster dynamics after photoexcitation.
  2. Part 2.2 is dedicated to the attribution of the different relaxation regimes observed in TA spectra. In order to not be lost by the variety of studied systems, this part could be split in 3 subsections labelled for instance (i) Monomer, (ii) impact of counterion and (iii) dimer.
  3. A schematic representation of the different involved processes in the relaxation mechanism for both monomer and dimer would really help the reader to have a global picture of the paper.

Additional suggestions and questions:

  1. Abstract:
    1. Perhaps start the abstract by an introducing sentence about the general context that justify this study, the role of counterion and dimerization.
    2. Line 16-18: In the sentence “The examination … photoexcitation.”, I suggest to add in the end: “, related to charge transfer to solvent and localized excited states in the cluster.”.
    3. Line 27-29: The sentence “The resulting transient absorption … can be observed.” Could be moved line 18 before enumerating the 4 different relaxation processes.
  1. Introduction:
    1. Line 37: Perhaps be more specific about research in chemistry. Applications in synthesis, photocatalysis, light-harvesting?
    2. Line 50: Can the author give more explanations why the NIR range is more adapted to probe excess e- formation while UV/Vis is more suiting for the cluster dynamics? Ideally, an energy diagram would really help to a global picture of the different processes. This diagram would echo the diagram I am proposing for discussion.
    3. Line 62-65: This sentence could go the discussion.
    4. Line 65-66: Why not saying explicitly here the impact of dimerization on the long-term dynamics?
    5. Line 71-72: Add in the end “ and the role of counterions and dimerization.”
  1. Part 2.1:
    1. Line 83: Could you give more details about the nature of calculations reported in ref 2?
    2. Line 83: Can you specify the energy range?
    3. Line 86: What is the origin of this resonance? Can it be resonance transfer between both clusters ro something similar to a plasmon resonance?
    4. Figure 2: The cluster concentration is mentioned here and in the experimental methods. Can we exclude formation of dimers in monomer solutions?
  1. Part 2.2:
    1. I suggest splitting it into 3 parts: (i) Monomer, (ii) impact of counterion, and (iii) dimer.
    2. Line 93: TA already defined before.
    3. Line 99: 440 nm or 400 nm (see page 5)?
    4. Figure 3: Please mention excitation wavelength. How the inverted ground state absorption spectrum is obtained? Perhaps it can be given in SI…
    5. Figure 4: Time axis break does not help to evidence the t3 Sentence line 110 could be moved in the text. According to the text, Fit are used to extract the timescale of the 4 different relaxation dynamics, it seems in contradiction with “To guide the eye, fit curves are displayed as lines.”
    6. Line 115-118: These sentences would be useful in SI file to describe the fit model.
    7. Line 119: After “Table 1”, I would add “for all compounds”. Why not justifying the need of 4 different decay times here?
    8. Line 119: Can you explain how to obtain decay associated spectra (DAS) and what they bring? For instance in SI?
    9. Line 128: Can you justify more why triplet states can not be populated here?
    10. Line 135: I would mention that it's concerning Ge9- clusters.
    11. Line 145: The absorption spectrum for Na0-e- pair could be presented in Fig S2.
    12. Line 161: What is ESI?
    13. Figure 6: In the caption Fig. S3 and S5 are mentioned. So why referring to Fig.S5 before S4?
    14. Line 188-191: This sentence is long and unclear, which does not allow to understand why the study of the dimer is necessary. In addition, the authors do not return to this point later.
    15. Line 192: Please remove Fig. 7 caption from the text.
  1. Discussion:
    1. Line 215: Here starts the global discussion. Giving a general picture of the different relaxation mechanisms for both monomer and dimer and summarize the role of counterions and dimerization would help the reader.
    2. Line 230: Why it does not make sense for monomers?
    3. Line 234-237: How quasi conduction band or excited state (100 ps) can compete with CTTS (1 ps), as they have very different time scales? I guess authors meant relaxation of solvated electrons!? A reference would be appreciated.
    4. Line 247-250: It is not clear to me why the trapping field of Zn atom is the main limiting factor for e- escape rather a faster charge transfer between both clusters.
    5. Line 255-258: These prospective suggestions could be moved to the conclusion.
  1. Part 3.3:
    1. Line 282: What is the energy used? I guess several µJ per pulse?
    2. Line 287: How sum-frequency mixing is done? With a BBO crystal?
    3. Line 294: TA already defined.
  1. Supplementary Information file:
    1. It should be improved by giving the main conclusions that can be done from these additional results rather than being a list of figures.
    2. I am wondering whether Fig S1 is useful. At least all curves should be on the same graph for easier comparison between Ge clusters.
    3. In Fig.S3, the name of cluster could be mentioned on each graph. Why the pump energy is different for the dimer? To be resonant with the absorption band at 388 nm?
    4. The fit function needs comments. What justifies this model?

Author Response

Response to referee 1

Response to the general comments: 

  1. In abstract/introduction, I suggest them to be more specific on the interest to study ultrafast dynamics in such systems in this context and why comparing monomer with counter ions and neutral dimer to understand cluster dynamics after photoexcitation.
    Response: We followed the suggestion and changed the abstract in lines 15-21 and the introduction starting at line 37.
  2. Part 2.2 is dedicated to the attribution of the different relaxation regimes observed in TA spectra. In order to not be lost by the variety of studied systems, this part could be split into 3 subsections labeled for instance (i) Monomer, (ii) impact of counterion, and (iii) dimer.
    Response: Thank you for this comment. We added the following headings for better overview: Line 103: “2.1 Dynamics of the monomer”, Line 180: 2.2.2 Impact of the counterion” and Line 213: “2.2.3 Dynamics of the dimer”
  3. A schematic representation of the different involved processes in the relaxation mechanism for both monomer and dimer would really help the reader to have a global picture of the paper.
    Response: We agree with the reviewer and insert reaction equations in lines 164-169 and a schematic representation to depict our results in line 268.

 Response to the additional suggestions and questions:

 Abstract:

  1. Perhaps start the abstract by introducing a sentence about the general context that justify this study, the role of counterion, and dimerization.
    Response: In line with this comment, we rewrote the sentences in line 14-21.
  2. Line 16-18: In the sentence “The examination … photoexcitation.”, I suggest to add in the end: “, related to charge transfer to solvent and localized excited states in the cluster.”.
    Response: We added in Line 18: “[…] photoexcitation related to charge transfer to solvent and localized excited states in the cluster.”
  3. Line 27-29: The sentence “The resulting transient absorption … can be observed.” Could be moved line 18 before enumerating the 4 different relaxation processes.
    Response: Thank you for this suggestion. We moved the sentence to line 19.

 Introduction:

  1. Line 37: Perhaps be more specific about research in chemistry. Applications in synthesis, photocatalysis, light-harvesting?
    Response: The reviewer highlighted an important point. We rewrote the introduction part starting at line 37.
  2. Line 50: Can the author give more explanations why the NIR range is more adapted to probe excess e- formation while UV/Vis is more suiting for the cluster dynamics? Ideally, an energy diagram would really help to a global picture of the different processes. This diagram would echo the diagram I am proposing for discussion.
    Response: We agree with the reviewer and explained in more detail starting at line 58 that the solvated electron spectrum in thf among other species is observable in the NIR spectral range.
  3. Line 62-65: This sentence could go the discussion.
    Response: It can now be found in lines 181-184.
  4. Line 65-66: Why not saying explicitly here the impact of dimerization on the long-term dynamics?
    Response: We changed lines 76-77: “Complementing studies on [(Hyp)3Ge9ZnGe9(Hyp)3][27, 33] (Ge9ZnGe9) show a strong influence of dimerization on the long timescale dynamics of the cluster which is crucial for the understanding of possible relaxation pathways.”
  5. Line 71-72: Add in the end “ and the role of counterions and dimerization.”
    Response: We thank the reviewer for the remark.

Part 2.1:

  1. Line 83: Could you give more details about the nature of the calculations reported in ref 2?
    Response: As suggested, we gave more details about the calculations in lines 90 and 94.
  2. Line 83: Can you specify the energy range?
    Response: Yes, we specified the energy range in line 92.
  3. Line 86: What is the origin of this resonance? Can it be resonance transfer between both clusters or something similar to a plasmon resonance?
    Response: Thank you for this question. These transitions can be found in the calculations as described in lines 94-97 but its physical origin remains obscure.
  4. Figure 2: The cluster concentration is mentioned here and in the experimental methods. Can we exclude the formation of dimers in monomer solutions?
    Response: Thank you for this important question. The concentration was about 1 mmol/L leading to an absorption spectrum which does not change with varying concentration. We added the concentration in line 100.

Part 2.2:

  1. I suggest splitting it into 3 parts: (i) Monomer, (ii) impact of counterion, and (iii) dimer.
    Response: In line with point 2. we split this part into three.
  2. Line 93: TA already defined before.
    Response: We thank the reviewer for this comment and corrected this mistake in line 105.
  3. Line 99: 440 nm or 400 nm (see page 5)?
    Response: We thank the reviewer for highlighting this misleading point. We rewrote lines 106-108. The excited-state absorption has a spectral broad featureless appearance, which is superimposed by the GSB around 400 nm. 440 nm is not the real maximum of the band and has been removed from the text.
  4. Figure 3: Please mention the excitation wavelength. How the inverted ground state absorption spectrum is obtained? Perhaps it can be given in SI…
    Response: Thank you for highlighting the missing of the excitation wavelength in line 113. We corrected the misleading wording: the absorption spectrum has just an inverted OD axis.
  5. Figure 4: Time axis break does not help to evidence the t3 Sentence line 110 could be moved in the text. According to the text, Fit are used to extract the timescale of the 4 different relaxation dynamics, it seems in contradiction with “To guide the eye, fit curves are displayed as lines.”
    Response: We are sorry for the wording here and removed the misleading part in Figure 4 in line 125. The time axis break was shifted to have a better look on the third time constant and we moved the sentence to line 118 in the text.
  6. Line 115-118: These sentences would be useful in SI file to describe the fit model.
    Response: We moved the sentences to the SI to explain the fit function.
  7. Line 119: After “Table 1”, I would add “for all compounds”. Why not justifying the need for 4 different decay times here?
    Response: We followed the reviewer and rewrote the sentences in lines 127-129. The four time constants were essential to fit the data adequately. Their interpretation is given throughout the discussion.
  8. Line 119: Can you explain how to obtain decay associated spectra (DAS) and what they bring? For instance in SI?
    Response: We added the following sentence in lines 130 and 131: “The DAS show the absorption spectra of the excited states, which correspond to the respective time constant.”
  9. Line 128: Can you justify more why triplet states can not be populated here?
    Response: We thank the reviewer to highlight this important point. Triplet state energies are too far away from the singlet energies and ISC is therefore unlikely. (Lines 139-141)
  10. Line 135: I would mention that it's concerning Ge9- clusters.
    Response:
  11. Line 145: The absorption spectrum for Na0-e- pair could be presented in Fig S2.
    Response: The [Na0,e-] pair spectrum is not depicted in reference 13 but it is described as two spectra: one of Na0 (we added it in Figure S2) and the solvated electron spectrum in thf.
  12. Line 161: What is ESI?
    Response: It means electronic supplementary information. We changed it to SI.
  13. Figure 6: In the caption Fig. S3 and S5 are mentioned. So why referring to Fig.S5 before S4?
    Response: We referred to Figures S 3, S 4, and S 5. For an easier view, we added in line 196: SI Figures S 3, S 4, and S 5.
  14. Line 188-191: This sentence is long and unclear, which does not allow us to understand why the study of the dimer is necessary. In addition, the authors do not return to this point later.
    Response: Thank you for highlighting this point. In lines 209-211 we rewrote the sentence and return to this point in line 239.
  15. Line 192: Please remove Fig. 7 captions from the text.
    Response: We are sorry for this formatting error and removed it in line 215.

Discussion:

  1. Line 215: Here starts the global discussion. Giving a general picture of the different relaxation mechanisms for both monomer and dimer and summarize the role of counterions and dimerization would help the reader.
    Response: We summarized shortly in lines 235-240 the results that we described so far.
  2. Line 230: Why it does not make sense for monomers?
    Response: The counterion is not directly linked to the monomer cluster. Therefore, an intramolecular charge transfer as known from the iron species is unlikely. We added some further explanation to this point in linen 256-257.
  3. Line 234-237: How quasi conduction band or excited state (100 ps) can compete with CTTS (1 ps), as they have very different time scales? I guess authors meant relaxation of solvated electrons!? A reference would be appreciated.
    Response: We thank the reviewer for highlighting this misleading part. The competing processes are the relaxation into the lower excited states and the CTTS which occur on the same timescale (<1 ps). We rewrote lines 261-264.
  4. Line 247-250: It is not clear to me why the trapping field of Zn atom is the main limiting factor for e- escape rather a faster charge transfer between both clusters.
    Response: Referring to the results from [Ge9(Hyp)3FeCp(CO)2] only one Ge9- entity is involved. The Zn-dimer shows similar transient absorption properties indicating a similar charge transfer behavior. (Line 281)
  5. Line 255-258: These prospective suggestions could be moved to the conclusion.
    Response: We moved them into the conclusion. (lines 379-384)

Part 3.3:

  1. Line 282: What is the energy used? I guess several µJ per pulse?
    Response: We used 2-3 µJ per pulse. (Line 341)
  2. Line 287: How sum-frequency mixing is done? With a BBO crystal?
    Response: We thank the reviewer for highlighting this missing information. We used a BBO crystal. (Line 347)
  3. Line 294: TA already defined.
    Response: We thank the reviewer for this comment and corrected this mistake in line 353.

Supplementary Information file:

  1. It should be improved by giving the main conclusions that can be done from these additional results rather than being a list of figures.
    Response: The figures given in the SI are of particular interest to strengthen the results and conclusions given by the figures/measurements in the main manuscript. We now added a short description for each figure in the SI.
  2. I am wondering whether Fig S1 is useful. At least all curves should be on the same graph for easier comparison between Ge clusters.
    Response: We think that absorption spectra are essential in such a study but we show now a graph with all spectra in Figure S 1.
  3. In Fig.S3, the name of the cluster could be mentioned on each graph. Why the pump energy is different for the dimer? To be resonant with the absorption band at 388 nm?
    Response: The spectra after 267/258 nm excitation of Ge9ZnGe9 as well as K(crypt)Ge9 can be found in the main manuscript. The reviewer is right, the additional excitation is to be resonant with the absorption band at 388 nm.
  4. The fit function needs comments. What justifies this model?
    Response: We thank the reviewer for this comment. We added further details about the fit function in the SI.

Reviewer 2 Report

This paper presents results from ultrafast spectroscopy experiments on a Ge9- with three different cations and a dimer cluster linked by a Zn atom.  The authors are interested in the electronic relaxation processes upon excitation by UV light.  The primary result is that the monomer cluster exhibits decay consistent with a charge transfer to solvent (THF in this case) while the dimer cluster appears to suppress it.  Other than that, the results appear to be "basically what you would expect." 

While this paper offers a credible explanation of the observed phenomena, it is unclear to me what the significance of these results are.  Particularly given that, for the most part, the relaxation mechanisms could be guessed ahead of time, it isn't clear to me why any reader who isn't specifically interested in these Ge clusters should need to know.  This is largely due to the terse introduction, which explains what the systems are but not why any reader who isn't already interested, should be.  In addition, the discussion is fairly dense, which may be inaccessible to a non-expert reader.  Some diagrams or cartoons would go a long way to help.  I'm not saying that it isn't possible that there is a reason a broad range of chemists should see this paper, just that I don't see it expressed in the paper as written.  As written, I think this paper could be appropriate for a specialized journal, but I would encourage the authors to expand the introduction.

Some specific critiques:

  1. There is almost no characterization data presented, and the discussion of the synthetic protocols is minimal.
  2. Much of the description contains statements like, "Obviously", "Apparently", and "Naturally" with limited detail as to how the data they presented led to the conclusion that they reached.  Similarly, there are overstatements such as "conclusive" that appear much stronger than justified.
  3. "crypt" is not defined until after its first use.
  4. More detail needs to be provided to be convincing that Ge9- is similar to Au25(SH)18-.
  5. What is the source of the structures presented in Figure 1?  Are they xtal structures or computed?
  6. Please explain the statement that Zn impacts the Franck Condon factor of some putative unidentified transition.

Author Response

Response to referee 2

  1. There is almost no characterization data presented, and the discussion of the synthetic protocols is minimal.
    Response: We thank the reviewer for this comment and added further details in the Materials and Methods part in lines 293-330.
  2. Much of the description contains statements like, "Obviously", "Apparently", and "Naturally" with limited detail as to how the data they presented led to the conclusion that they reached.  Similarly, there are overstatements such as "conclusive" that appear much stronger than justified.
    Response: We removed the unneeded statements and attenuate the overstatements, for example, lines 119, 203, 205, and 244.
  3. "crypt" is not defined until after its first use.
    Response: We thank the reviewer for this comment and added the definition in line 75.
  4. More detail needs to be provided to be convincing that Ge9- is similar to Au25(SH)18-.
    Response: We thank the reviewer for highlighting this point. The comparison with the gold cluster is not needed in this context and was therefore removed from line 94.
  5. What is the source of the structures presented in Figure 1?  Are they xtal structures or computed?
    Response: The structures in Figure 1 are from crystallographic data. We added this in line 71.
  6. Please explain the statement that Zn impacts the Franck Condon factor of some putative unidentified transition.
    Response: In line with the response to referee 1 we rewrote this part in the manuscript in lines 94-97. Despite calculations, the assignment is still not clear.

Reviewer 3 Report

Although I cannot comment on the experimental details of their study because I am not a specialist, the paper is well organized, and their analysis, based on which they propose four processes of the cluster dynamics, seems reasonable.
There are several points to be improved.
Figure 2 is not explicitly referred to in the text.
In the description of Fit function in SI, the authors should provide a more clear explanation. It is sure that the equation expresses the superposition of n decay modes. But the physical meaning and/or the theoretical ground for the erf term in the right side of the equation.
In addition, this might be too application-oriented for this study, but if they touched the relation to other topics such as functional molecules exhibiting photo induced electron transfer, the paper would be more intriguing.

Author Response

Response to referee 3

  1. Figure 2 is not explicitly referred to in the text.
    Response: We referred to Figure 2 in line 86.
  2. In the description of Fit function in SI, the authors should provide a more clear explanation. It is sure that the equation expresses the superposition of n decay modes. But the physical meaning and/or the theoretical ground for the erf term in the right side of the equation.
    Response: In line with the response to referee 1 we added a paragraph in the SI to describe the fitting procedure. The error function is a mathematic approximation to deconvolve the steplike rise of the signal that originates from two Gaussian pulse envelopes (pump and probe pulses).
  3. In addition, this might be too application-oriented for this study, but if they touched the relation to other topics such as functional molecules exhibiting photo induced electron transfer, the paper would be more intriguing.
    Response: We agree with the reviewer and rewrote in line with referees 1 and 2 the introduction part starting at line 37.